# An Early Miocene kentriodontoid (Cetacea: Odontoceti) from the western North Pacific, and its implications for their phylogeny and paleobiogeography

**Zixuan Guo** [1]*, **Naoki Kohno** [1,2]

1 Graduate School of Life and Environmental Sciences, University of Tsukuba, Tsukuba, Japan,
2 Department of Geology and Paleontology, National Museum of Nature and Science, Tsukuba, Japan

* guo_z@geol.tsukuba.ac.jp

**Data Availability Statement:** All relevant data are within the paper and its Supporting Information files.

## Abstract

So–called 'kentriodontids' are extinct dolphin–like odontocetes known from the Early to Late Miocene worldwide. Although recent studies have proposed that they were monophyletic, their taxonomic relationships still remain controversial. Such a controversy exists partly because of the predominance of primitive morphologies in this taxon, but the fact is that quite a few 'kentriodontids' are known only from fragmentary skulls and/or isolated periotics. A new 'kentriodontid' *Platysvercus ugonis* gen. et sp. nov. is described based on a nearly complete skull from the upper Lower Miocene Sugota Formation, Akita Prefecture, northern Japan. Based on the phylogenetic analysis of *P. ugonis* described here, the monophyly of the 'kentriodontids' is confirmed, and it is recognized as the superfamily Kentriodontoidea. This new superfamily is subdivided into two families as new ranks: Kentriodontidae and Lophocetidae. Based on the paleobiogeographic analysis of the Kentriodontoidea, their common ancestor emerged in the North Pacific Ocean and spread over the Northern Hemisphere. Initial diversification of the Kentriodontidae in the North Pacific Ocean and the Lophocetidae in the North Atlantic Ocean was recognized as a vicariance event. The diversification and extinction of the Kentriodontoidea could have been synchronously influenced by climate events during the Middle Miocene.

## Introduction

So-called 'kentriodontids' are an extinct group of odontocetes known from the Early to Late Miocene worldwide. At least 14 genera and 26 species are described as 'kentriodontids,' and they are sometimes recognized as polyphyletic stem taxa within the Delphinida, including the Delphinoidea and Inioidea [1–5]. Although many studies have concentrated to understand their inter–generic and inter–specific relationships within the Delphinida, some of previously described specimens of 'kentriodontids' were fragmentary, and they have sometimes been classified as paraphyletic and/or unresolved taxa. Some recent studies [5–9] attempted to solve this problem. In particular, some of these studies [8,9] supported the monophyletic relationships of

**Funding:** The authors received no specific funding for this work.

**Competing interests:** The authors have declared that no competing interests exist.

all previously known 'kentriodontids', which comprise the sister group of the Dephinoidea, while other studies [5–7] show quite different, polyphyletic topologies among them. Moreover, quite a few new species of 'kentriodontids' have been reported in the past decades. Therefore, a more comprehensive study for 'kentriodontids,' especially their phylogenetic relationships, is necessary to refine the systematics of 'kentriodontids.'

In addition to the phylogenetic relationships, previous studies on 'kentriodontids' suggested that their ecological niche was thought to be similar to the extant delphinoids [10,11]. They show high diversity through the late Early and early Late Miocene, and their high diversity during those periods is thought to be parallel to the present day Delphinoidea. In particular, the Delphinoidea originated from the Late Miocene [12,13], and seemed to be replaced by the kentriodontids afterward, while this turnover event also remains unexplained. Furthermore, many geological events occurred during the Miocene, with notable changes in climate change (for example, the Miocene Climatic Optimum and Middle Miocene Climate Transition [14], tectonics [15], ocean circulation [16], and ice–sheet volume [17]). However, only a few studies have investigated the effects of these events on marine vertebrates [18]. Accordingly, their geologic ages and geographic distribution are important to understand the pattern and process of the evolution and diversification of 'kentriodontids.' In this context, dense taxon sampling is necessary because, for instance, only two 'kentriodontid' species [7,19] have been known from the western North Pacific.

Here we describe a new 'kentriodontid' skull from the upper Lower Miocene of Japan as an addition to the western North Pacific record of 'kentriodontids' (Fig 1). Also, we determine its position in the crown Odontoceti and re-examine 'kentriodontid' phylogeny with further species added. We then try to revise their systematics and reconstruct their paleobiogeography based on our rigorous analysis, considering the evolution and diversification of 'kentriodontids'.

## Materials and methods

### Phylogenetic analysis

The phylogenetic position of the new specimen (i.e. UTHFM 00034) was analyzed using the data matrix of Guo and Kohno [8], which was modified in minor ways (see Supporting Information). In addition to the 'kentriodontid' taxa that were used for the phylogenetic analysis in Guo and Kohno [8], three previously known 'kentriodontids' (i.e., *Kentriodon hobetsu* from the Middle Miocene of Japan, *Kentriodon hoepfneri* from the Middle/Upper Miocene of Germany and *Sophianacetus commenticius* from the Middle Miocene of Hungary) were included in our analysis with UTHFM 00034. The character matrix and OTUs analyzed by Guo and Kohno [8] are a combination of Tanaka et al. [20] and Kimura and Hasegawa [7], which were derived from those prepared by Peredo et al. [5], Lambert et al. [6], Tanaka and Fordyce [21], Geisler et al. [22], Murakami et al. [12], and Geisler et al. [23]. Our data matrix includes almost all the 'kentriodontids' that are known at least by crania. The analysis performed here was based on 107 taxa and 387 morphological characters, with a backbone constraint of cetaceans [24–26] (see Supporting Information). In total, we included 21 previously known 'kentriodontid' taxa with UTHFM 00034 into our phylogenetic analysis (see Supporting Information).

Phylogenetic analysis was conducted with TNT 1.5 [27,28], using the 'New Technology Search' with default values, found minimum length trees 1,000 times with a backbone constraint from extant taxa [24–26], as mentioned above. All characters were treated as unweighted and unordered. To measure the stability of each node, we estimated Bremer support with score up to 30 worse than best (in 30 searches, each 1 worse than previous) and Standard Bootstrap support with 1000 replicates for the consensus tree.

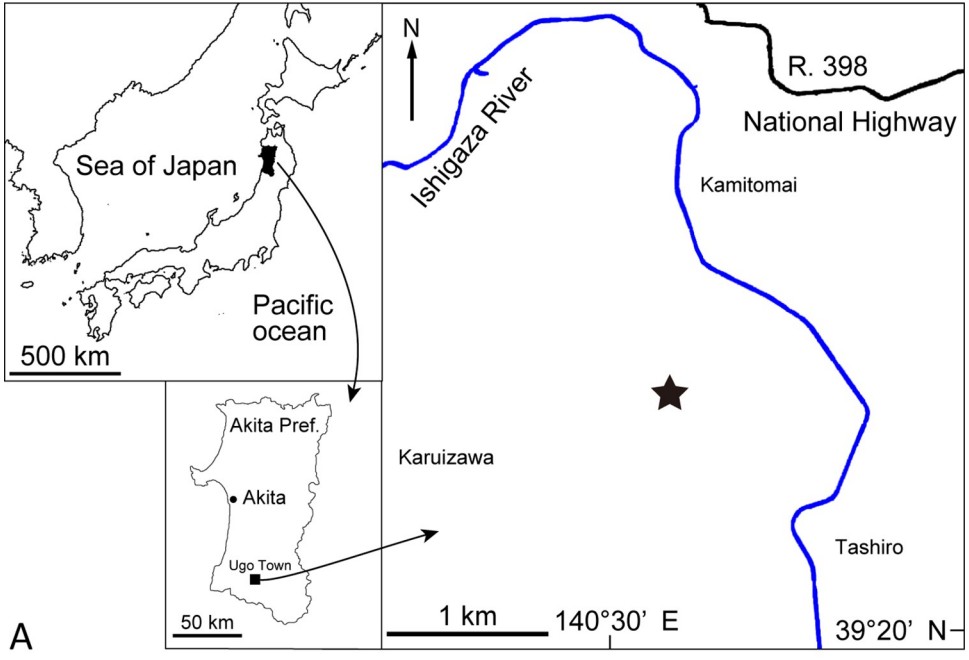

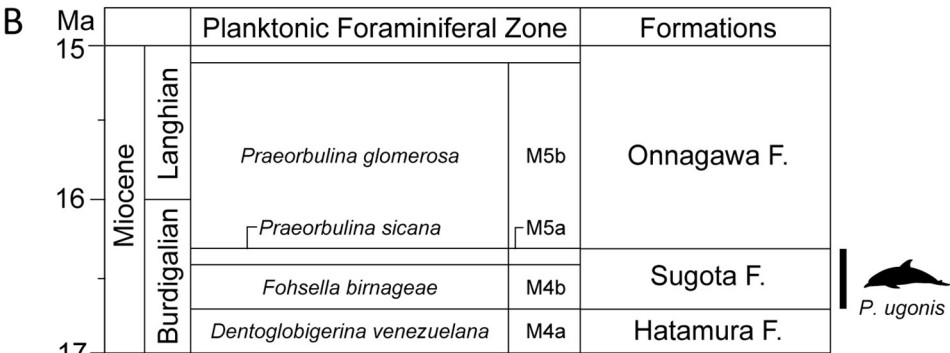

**Fig 1. Geographical and geological contexts of the type locality of *Platysvercus ugonis* gen. et sp. nov.** (A) the type locality of *Platysvercus ugonis* gen. et sp. nov., holotype, UTHFM 00034, modified from the Topographic Maps from The Geospatial Information Authority of Japan (GSI). (B) planktonic foraminiferal zone and stratigraphic diagram, modified from Ozawa [40] and Wade et al. [39].

### Ancestral range reconstruction analysis

Ancestral range reconstructions using the Statistical Dispersal–Vicariance Analysis (S–DIVA) [29], implemented in the software RASP 4 [30,31], were performed to understand the paleobiogeographic distributions of 'kentriodontids.' The S–DIVA is a parsimony–based method to reconstruct the frequency of alternative ancestral distribution at each node in the phylogeny. S–DIVA considers vicariance events as carrying a lower cost than dispersal and extinction events, and then accounts for all possible ancestral distributions and calculates the probabilities.

All the taxa within the most parsimonious trees were used in this analysis. The condensed tree was used by the user–specified consensus tree; i.e., the strict consensus tree from the phylogenetic analysis. Due to the distribution of marine mammals that are usually considerably broader than terrestrial animals with ambiguous distribution boundaries, we assigned seven distributional regions of the oceans as follows: (A) North Pacific Ocean, (B) North Atlantic

Ocean, (C) South Pacific Ocean, (D) South Atlantic Ocean, (E) Indian Ocean, (F) Arctic Ocean, and (G) Southern Ocean, for both extinct and extant species of odontocetes. The distributional information for extant taxa was collected from the IUCN Red List of Threatened Species [32], and the extinct taxa were assigned based on the Paleobiology Database (PBDB, www.paleobiodb.org). The S–DIVA analysis was calculated with the following parameters: max areas at each node = 4; allow reconstruction with max reconstructions = 100; max reconstructions for final tree = 1000.

## Institutional abbreviations

UTHFM, Ugo Town History and Folklore Museum, Ogachi County, Akita, Japan; NMNS–PV, fossil vertebrate collections at the Department of Geology and Paleontology, National Museum of Nature and Science, Tsukuba, Japan; NSMT–M, Mammal Collections at the Department of Zoology, the National Museum of Nature and Science, Tsukuba, Japan.

## Anatomical terminology

The skull terminology follows Mead and Fordyce [33] and Ichishima [34].

## Nomenclatural acts

The electronic edition of this article conforms to the requirements of the amended International Code of Zoological Nomenclature, and hence the new names contained herein are available under that Code from the electronic edition of this article. This published work and the nomenclatural acts it contains have been registered in ZooBank, the online registration system for the ICZN. The ZooBank LSIDs (Life Science Identifiers) can be resolved and the associated information viewed through any standard web browser by appending the LSID to the prefix ""http://zoobank.org/"". The LSID for this publication is: urn:lsid:zoobank.org:pub:B46B6775-A7E2-4081-97D9-1D4906D0E3B4. The electronic edition of this work was published in a journal with an ISSN, and has been archived and is available from the following digital repositories: PubMed Central, LOCKSS.

## Ethics statement

Ethics approval is not applicable, because this article does not contain any studies with human or animal subjects.

## Systematic paleontology

ARTIODACTYLA Owen, 1848
 CETACEA Brisson, 1762
 ODONTOCETI Flower, 1867
 DELPHINIDA Muizon, 1984
 Superfamily KENTRIODONTOIDEA (Slijper, 1936) new rank

### Diagnosis of superfamily

It differs from other superfamilies of the Delphinida in having the following derived characters: the premaxillae at anterior of the rostrum mediolaterally compressed (character 3), the mesorostral groove posteriorly constricted, anteriorly abruptly diverges at the level anterior to the nares and behind the level of the antorbital notch (character 7), the lacrimal or the jugal forms the dorsolateral edge of internal opening of the infraorbital foramen (character 56), the infratemporal crest forms a curved ridge (character 62), the dorsal margin of the

mesethmoid low (character 104), the alisphenoid laterally exposed broadly in the temporal fossa (character 157), the basioccipital crests angle 15˚–40˚ in the ventral view (character 225), the hypoglossal foramen separated by a thick bone from the jugular foramen (character 227), the pars cochlearis ventrolaterally convex (character 278), the basihyal fused with the thyrohyal (character 326), and the transverse processes of lumbar vertebrae angle 45˚ or more (character 328).

**Type family.** Kentriodontidae Slijper, 1936.

**Included families.** Kentriodontidae Slijper, 1936 sensu Peredo et al., 2019 and Lophocetidae (Barnes, 1978) new rank with new definition

**Etymology of superfamily.** The superfamily name is from the type family, Kentriodontidae.

Family KENTRIODONTIDAE (sensu Peredo et al., 2018) with new definition

**Diagnosis of family.** It has the following derived characters: the pterygoids partially contacting each other in the anteroventral view (Character 183), the anterior process of periotic contact with the tympanic bulla anterior to the accessory ossicle but no clear fossa for the articulation on the periotic (Character 245), the foramen singulare separated by the equal height partitions from the spiral cribiform tract and the proximal opening of the facial nerve canal (Character 264), the proximal opening of the facial nerve canal continuous with an anterior fissure (Character 266), and the angle between the anterior process of periotic and the anterior edge of pars cochlearis nearly 90 degrees, rectangular or semicircular pars cochlearis in ventral view (Character 277).

Family LOPHOCETIDAE (Barnes, 1978) new rank with new definition

**Diagnosis of family.** It differs from Kentriodontidae in having the following derived characters: 24–27 mandibular teeth (character 37), the premaxillary foramen located medially (character 70), a distinctly inflated process of the nasal at its anterolateral corner present, which tapers in width anteriorly and ventrally, and the angle between the dorsal and anterior face of the process rounded over (character 133), the posterior part of the suture between the nasals marked by a deep cleft (character 134), the postglenoid process ventrally tapers (character 169), the stylomastoid fossa situated on the posterior face of the pars cochlearis, posterodorsal to the stapedial muscle fossa (character 258), the dorsal edge of the tegmen tympani lateral to the aperture for the vestibular aqueduct presents as a faint ridge (character 273), and the deltopectoral tuberosity of humerus proximodistally centered (character 355).

**Type genus.** *Lophocetus* Cope, 1868

**Included genera.** *Atocetus* de Muizon, 1988, *Delphinodon* Leidy, 1869, *Hadrodelphis* Kellogg, 1966, *Lophocetus* Cope, 1868, *Liolithax* Kellogg, 1931, *Macrokentriodon* Dawson, 1996, *Tagicetus* Lambert et al., 2005, and *Platysvercus* gen. nov.

Genus *Platysvercus* gen. nov.

*LSID*. urn:lsid:zoobank.org:act:534A947D-EACC-406A-A853-D9D6041ADEE4

**Type and only species.** *Platysvercus ugonis*, sp. nov.

**Diagnosis of genus.** As for the type and only included species.

**Etymology of genus.** The genus *Platysvercus* is from Greek Platys, for broad or wide, and Greek Sverkos, for the back of the neck, referring to the transversely wide nuchal crest of the new genus

*Platysvercus ugonis* sp. nov.

(Figs 2–6; Table 1)

*LSID*. urn:lsid:zoobank.org:act:141138C8-3D58-4852-80AD-3A5E8237CCD2

**Holotype.** UTHFM 00034, a nearly complete skull, discovered and collected by Takashi Sato in June 1992.

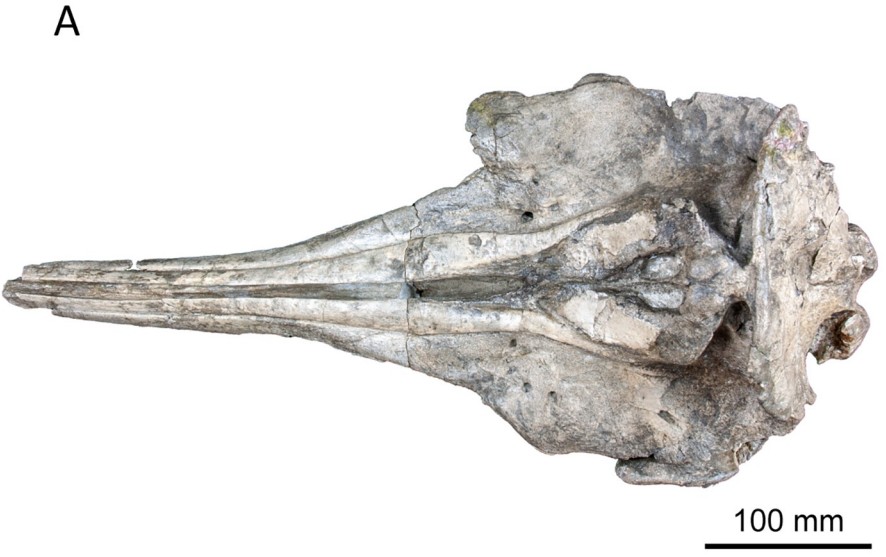

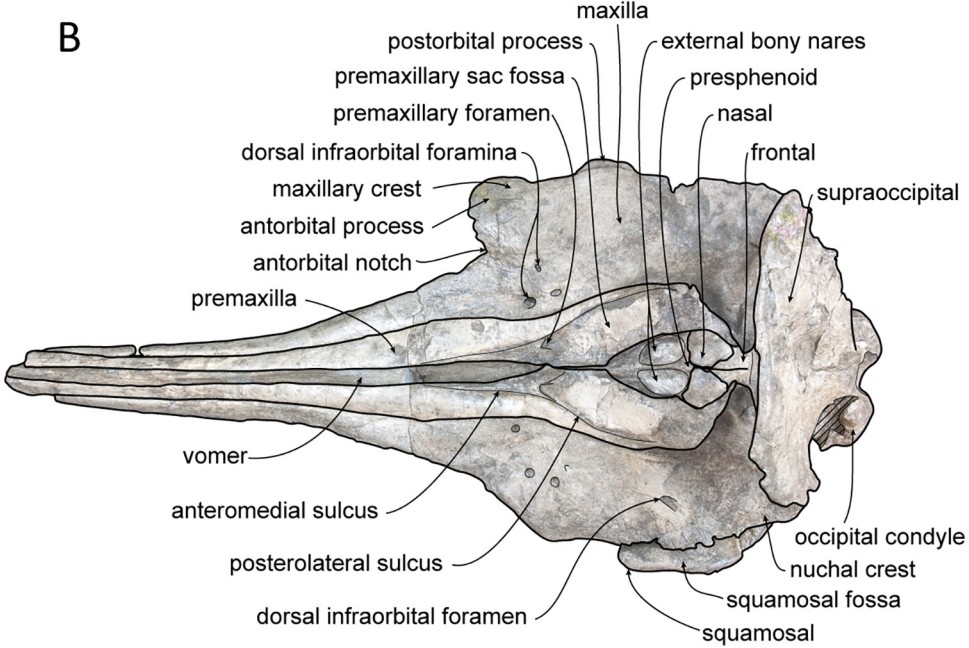

**Fig 2. Dorsal view of the skull of *Platysvercus ugonis* gen. et sp. nov., holotype, UTHFM 00034.** (A) photo. (B) corresponding line drawing with anatomical interpretations. Scale bar equals 100 mm.

**Diagnosis of Species.** It differs from other species within the family in having a unique combination of the characters: the premaxillary foramen located laterally from the midpoint to the premaxilla (character 70), the premaxillae raised up towards the midline but concave transversely between the anteromedial sulci emanating from the premaxillary foramen (character 121), and the anterior margin of the pterygoid sinus fossa located posterior to the antorbital notch (character 190).

A

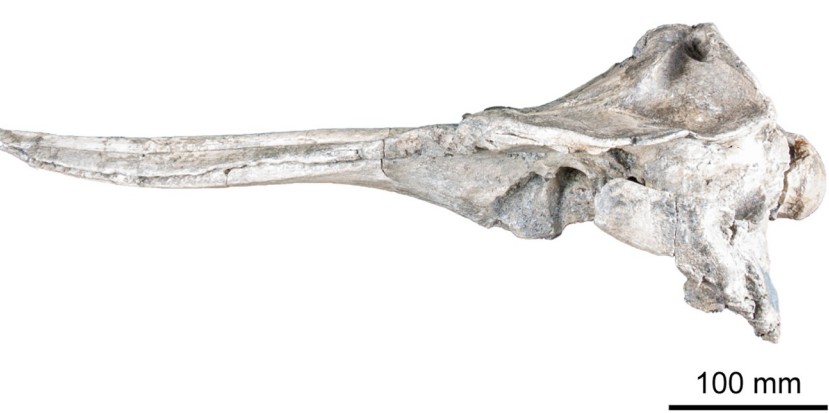

100 mm

B

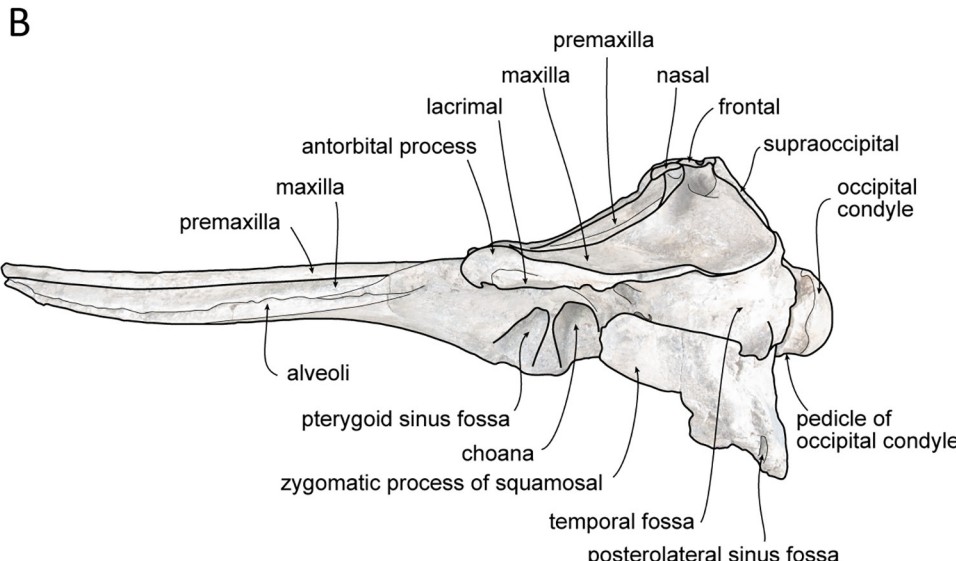

**Fig 3. Left lateral view of the skull of *Platysvercus ugonis* gen. et sp. nov., holotype, UTHFM 00034.** (A) photo. (B) corresponding line drawing with anatomical interpretations. Scale bar equals 100 mm.

**Etymology of species.**   The species name *ugonis* is derived from Ugo Town, where the holotype was discovered and collected.

**Type locality.**   The holotype was collected from a road construction site in Tashiro district, Ugo Town, Ogachi County, Akita Prefecture, Japan (39˚21'57"N, 140˚30'55"E; Fig 1).

**Formation and age.**   Sugota Formation, Early Miocene in age. The holotype was collected from the lowest part of the outcrop along the roadside during road construction. Judging from the stratigraphic horizon at the outcrop, this specimen was derived from the main part of the Sugota Formation. According to Sato et al. [35], Iwasa and Kikuchi [36], Fujioka et al. [37],

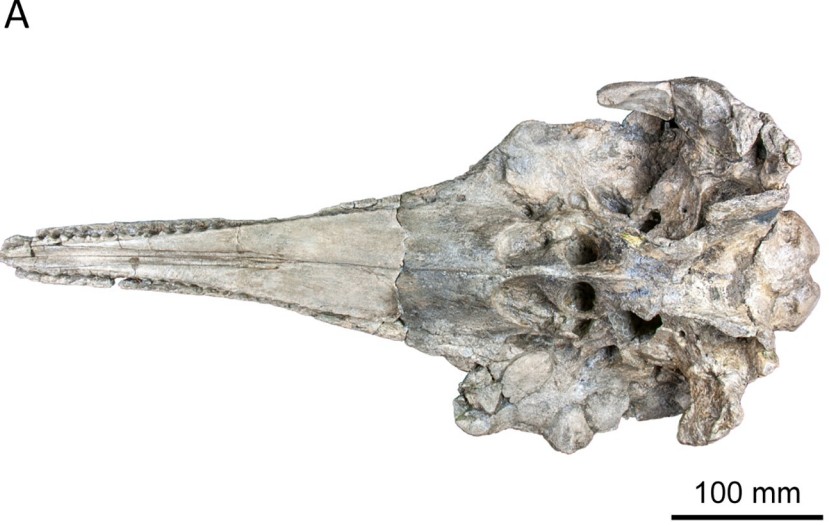

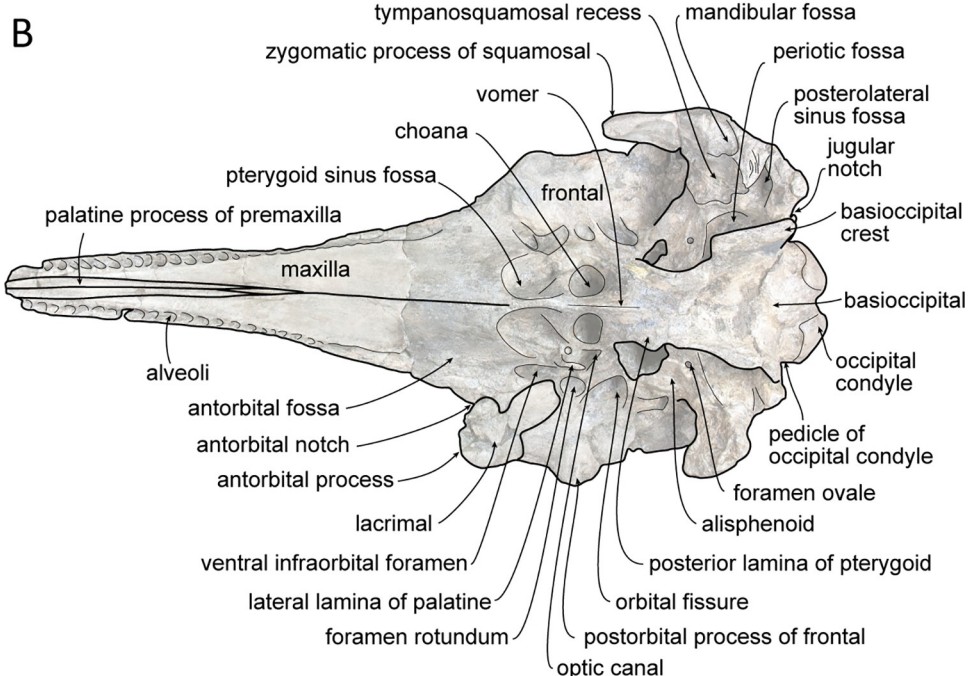

**Fig 4. Ventral view of the skull of *Platysvercus ugonis* gen. et sp. nov., holotype, UTHFM 00034.** (A) photo. (B) corresponding line drawing with anatomical interpretations. Scale bar equals 100 mm.

and Aita [38], the main part of the Sugota Formation is correlated with the NN4–NN5 calcareous nannofossil zone and N9–N10 planktonic foraminiferal zone. Also, according to Aita [38], the Sugota Formation has produced *Praeorbulina sicana* and *Fohsella birnageae*, which is correlated with the M4b–M5a planktonic foraminiferal zone [39], ranging in age between 16.70 and 16.29 Ma, late Burdigalian, late Early Miocene. The Sugota Formation is mainly composed

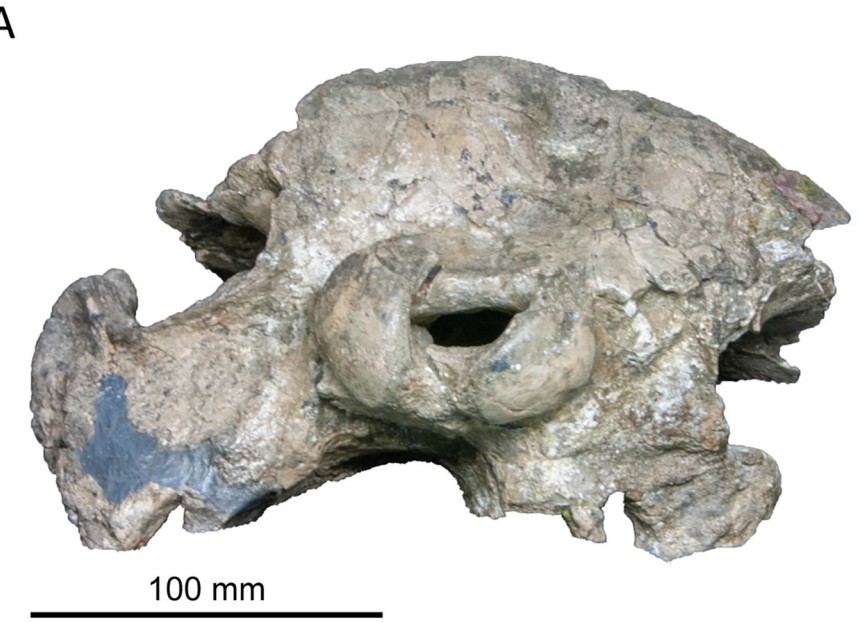

**Fig 5. Posterior view of the skull of *Platysvercus ugonis* gen. et sp. nov., holotype, UTHFM 00034.** (A) photo. (B) corresponding line drawing with anatomical interpretations. Scale bar equals 100 mm.

of sandstone with abundant mollusks [40]. Further, some *Desmostylus* teeth are known from this Formation [41].

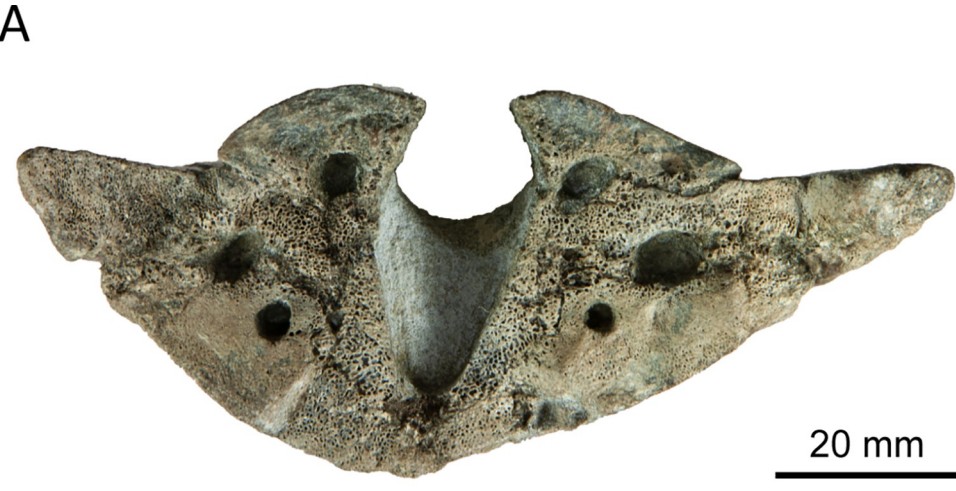

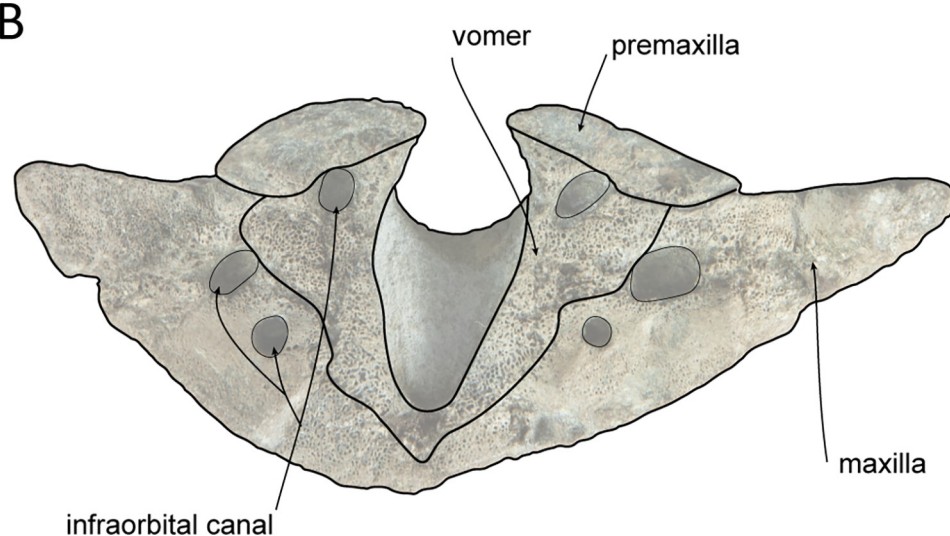

**Fig 6. Posterior view of the cross section of rostrum _Platysvercus ugonis_ gen. et sp. nov., holotype, UTHFM 00034, 50 mm anterior to the antorbital notch.** (A) photo. (B) corresponding line drawing with anatomical interpretations. Scale bar equals 20 mm.

## Description

**Cranium.** The cranium is nearly complete, and only the left antorbital notch, the right glenoid process of the squamosal, and most of the pterygoid are missing. Although the apex of the rostrum was slightly broken, it is clearly very long. Deformation is observed at the neurocranium, temporal fossa, and occipital condyle. In the posterior view, the cranium is deformed; the higher part of the cranium is turned clockwise and skewed to the right, and the posteroventral part of the cranium is higher on the left side than the right. The nasals and the premaxillae

**Table 1. Measurements (in mm) for the skull of *Platysvercus ugonis* gen et sp. nov., holotype, UTHFM 00034.**

| Dimension | Measurement |
| --- | --- |
| Condylobasal length, from tip of rostrum to hindmost margin of occipital condyles | 566+ |
| Length of rostrum, from tip to line across hindmost limits of antorbital notches | 318.5+ |
| Width of rostrum at base, along line across hindmost limits of antorbital notches | 152e |
| Width of rostrum at 60 mm anterior to line across hindmost limits of antorbital notches | 89.3 |
| Width of rostrum at mid–length | 54+ |
| Width of premaxillaries at mid–length of rostrum | 30.1 |
| Width of rostrum at 3/4 length, measured from posterior end | 49 |
| Distance from tip of rostrum to external nares (to mesial end of anterior margin of right naris) | 414+ |
| Distance from tip of rostrum to internal nares (to mesial end of posterior margin of right pterygoid) | 382+ |
| Distance between first posterior alveolus and antorbital notch | 58.9+ |
| Distance between anterior tips of right and left premaxillary foramina | 26.7 |
| Projection of premaxillae beyond maxillae measured from tip of rostrum to line across foremost tips of maxillae visible in dorsal view | 12.5+ |
| Width of premaxillae at posterior extremity | 49.6 |
| Maximum width of mesorostral groove on the rostrum | 12.5 |
| Width of mesorostral groove at rostrum base | 12.5 |
| Width of right premaxillary sac fossa | 37 |
| Width of left premaxillary sac fossa | 36.6 |
| Length of medial suture of nasals | 4.7 |
| Width of nasal bones | 43.3 |
| Maximum length of frontal at the vertex | 24.4 |
| Greatest preorbital width | 235.8e |
| Greatest postorbital width | 261.8e |
| Least supraorbital width | 224.2e |
| Greatest width of external nares | 36.1 |
| Greatest width across zygomatic processes of squamosals | 237.6e |
| Greatest width of premaxillae | 96 |
| Greatest parietal width | 181.6 |
| Greatest length of temporal fossa, measured to external margin of temporal crest | 144.3 |
| Greatest width of temporal fossa perpendicular to greatest length | 43.6 |
| Major diameter of temporal fossa proper | 52.1 |
| Minor diameter of temporal fossa proper | 82.9+ |
| Distance from foremost end of junction between nasals to hindmost point of margin of nuchal crest | 41.1 |
| Length of orbit–from apex of preorbital process of frontal to apex of postorbital process | 57 |
| Length of antorbital process of lacrimal | 31.7+ |
| Greatest width of internal nares | 54.2+ |
| Width across occipital condyles | 84.2 |
| Width of the foramen magnum | 32.3 |
| Height of right occipital condyle | 54.5 |
| Width of right occipital condyle | 31.8 |
| Maximum distance between basioccipital crests | 102.3+ |

Abbreviations: e, estimate; +, not complete.

are symmetrical in the dorsal view, while the vertex is slightly skewed right to the midline of the foramen magnum because of this deformation. The temporal fossa is dorsoventrally narrow in lateral view. The dorsal wall of the temporal fossa is low and flat, it is almost horizontal at the level of the orbital fossa, only shows slight posterior elevation.

**Premaxilla.**   The premaxillae are well preserved, and only the anterior tips are missing. The dorsal surface of the premaxilla is flat along the anterior rostral portion, while the flat area is rotated to face dorsolaterally, and its medial margin makes a sharp keel dorsally. The mesorostral groove opens along the medial margins of the premaxillae, starting from the level of the prenarial triangle, just anterior to the premaxillary foramen. In the anterior and posterior views of the cross sections of the rostral base (Fig 6), the mesorostral groove is U–shaped, and the medial edge of the premaxilla is slightly extended dorsomedially. The width of the premaxilla is narrow anteriorly and gradually widens posteriorly to the level of the prenarial triangle. Medially, the premaxillae do not cover the external bony nares and the medial border of the premaxillae are oriented following the anterolateral wall of the external bony nares. In dorsal view, the external bony nares are cordiform, formed anterolaterally by the premaxillae and posteriorly by the nasals and the presphenoid. The premaxillary foramen is positioned at the midpoint between the lateral and medial margins of the premaxilla and is located at the anterior margin of the premaxillary sac fossa and medial to the anteromedial sulcus. The premaxillary sac fossa is flat and gradually elevated posteriorly. Medially, it is higher than the portion lateral to the posterolateral sulcus of the premaxilla. The premaxilla contacts the nasal posteriorly, and the posterior end of the premaxilla is wedge–shaped. The premaxillae are symmetrical at the vertex; however, at the level just anterior to the prenarial triangle, the right premaxilla is slightly wider than the left. The posterior margin of the premaxilla is distinctly higher than that of the maxilla in lateral view.

**Maxilla.**   The maxilla is well preserved, and only the anterior tip of the rostrum and the left antorbital process are missing. In dorsal view, the maxilla is exposed along the rostrum, and it is almost the same width as the premaxilla. The lateral margin of the maxilla is slightly concave laterally. The surface of the maxilla faces dorsally at the rostral base, whereas it faces dorsolaterally toward the anterior tip of the rostrum, as is also the case for the premaxilla. In the broken section at the rostral base (Fig 6), the ventral surface of the maxilla is convex, while the dorsal surface is flat or slightly concave. Three infraorbital canals are preserved in the broken section, one positioned in the suture between the premaxilla and the vomer, and two in the maxilla, where they are closely located to the maxilla–vomer suture.

The antorbital notch is triangular and posteriorly deep. Although the anterior edge is slightly damaged, the shape of the right antorbital process is somewhat rectangular in dorsal view. There are four anterior and one posterior dorsal infraorbital foramina; the anterior dorsal infraorbital foramina are located at the level of the premaxillary foramen of the premaxilla, and the posterior dorsal infraorbital foramen is situated at the level of the external bony nares on the left maxilla while absent on the right one. The dorsal surface of the maxilla over the supraorbital process is flat. The maxillary crest is poorly defined posterior to the antorbital process. The facial fossa is shallow. The maxilla overlaps the frontal completely, and only part of the postorbital process can be observed in dorsal view. Posteriorly, the maxilla is somewhat square in dorsal view. The maxilla contacts the supraoccipital posteriorly, and its posterolateral margin reaches the anteroposterior level of the jugular notch of the exoccipital. The nuchal crest is straight and forms a frontal–maxillary suture. At the vertex, the maxilla is lower than the posterior end of the premaxilla, nasal, frontal, and nuchal crests. It lies anterior to the nuchal crest and forms a deep fossa lateral to the frontal and posterior to the nasal and premaxilla. In lateral view, the lateral margin of the maxilla from the level of the antorbital notch to the level of the ascending process is thin and flat.

In ventral view, the left and right maxillae contact each other at the midline, but they are not fused at the posterior half of the rostrum, while the maxilla is separated by the exposure of the palatine process of the premaxilla along the anterior half of the rostrum. Both sides of the maxilla preserve 28 alveoli, and the total length of the maxillary alveolar groove is 270 mm.

Each alveolus is small and almost the same size (4–6 mm in transverse width) from the apex to the posterior part of the alveoli. The maxillary–palatine suture cannot be defined. The anterior sinus fossa is shallow, and located at the same level or slightly posterior to the antorbital notch. Posterior to the anterior sinus fossa, the ventral infraorbital foramen is positioned immediately lateral to the pterygoid fossa, while the surrounding bones are not clear because the pterygoid and lacrimal are damaged.

**Palatine–pterygoid.** The maxillary–palatine suture is not preserved, and the anterior margin of the palatine is unclear. A fragment of the lateral lamina of the palatine is preserved and located immediately anterolateral to the external bony nares, but it is limited by the lateral width of the pterygoid sinus fossa. Most parts of the pterygoid are missing, and it only preserves its posterior lamina beside the vomer, although its suture with the vomer is unclear. The pterygoid sinus fossa is short. Its anterior wall is rounded and is extended posterior to the antorbital notch. The choanae are narrower than the pterygoid sinus fossa.

**Vomer.** The external bony nares are still filled with hard matrix, so the vomer is not observable in dorsal view. In the cross section of the rostral base, the dorsal surface of the vomer is U–shaped (Fig 2). It contacts the premaxilla dorsally and is surrounded by the maxilla ventrolaterally. In ventral view, the vomer forms the medial wall of the choanae. Posteriorly, the vomer–basisphenoid suture is not clearly defined.

**Presphenoid.** The nasal septum is low, and it contacts the presphenoid posteriorly. The nasal septum is dorsally lower than the dorsal surface of the surrounding premaxilla by 15 mm. Posteriorly, the presphenoid is high and is in contact with the nasal. In dorsal view, the presphenoid is convex and triangular in shape. It forms a sharp keel anterodorsally (11 mm in anteroposterior length), while the transverse width (21 mm) is narrow.

**Nasal.** The nasal lies just behind the external bony nares and is surrounded laterally by the premaxilla and posteriorly by the frontal. In dorsal view, the left and right nasals are symmetrical. The shape of each nasal is somewhat triangular in outline and pointed posteriorly towards its midpoint. Medially, the nasal contact withs each other and forms a short suture. The nasal reaches maximum width at the anterior margin, and it is slightly wider than the external bony nares. The dorsal surface of the nasal is smooth and only slightly concave anteromedially. Both the anterodorsal crest and the internasal fossa are not distinct. In anterodorsal view, the internal surface of the nasal is narrow and short, and it only extends ventrally from the anterolateral edge.

**Frontal.** The frontal is almost fully covered by the maxilla and only exposed in dorsal view at the vertex and the lateral part of the supraorbital. At the skull vertex, the frontal contacts the nasal anteriorly and the nuchal crest of the supraoccipital posteriorly. The frontal is anteriorly pointed and posteriorly pinched by the maxilla, while it widens again and contacts the supraoccipital at the posterior–most margin. In lateral view, the frontal at the vertex is lower than the nuchal crest and slightly lower than the nasal crest. The left and right frontal sutures can be observed in the dorsal view at the anterior part of the frontal, and it continues to the internasal suture.

The orbit is dorsoventrally located along the lateral edge of the posterior rostrum. It is low and shallow dorsoventrally. The preorbital process is convex ventrally and covered with a partially preserved lacrimal. There is no ventral orbital crest. The postorbital process is poorly preserved on the right side, and it only remains as a blunt infratemporal crest at the base. It is located at the level just anterior to the choana and opened dorsolaterally. The optic canal is only be distinct on the right side; it is located just lateral to the choana. The orbital fissure is located posteromedial to the postorbital process. Extending from the orbital fissure, the frontal groove is deep and wide (18 mm), and its anterior edge makes contact with the infratemporal crest. The foramen rotundum is positioned just anterior to the orbital fissure and is narrower than the latter.

**Lacrimojugal complex.** The portion of the left antorbital process is damaged and only remains an unclear suture of the lacrimal. The anterior edge of the right antorbital process of the lacrimal, which is also slightly damaged, is projected anterolaterally, and remains as a fragment beneath the process. It is thin in the supraorbital area and extends posteriorly to the anteroposterior level of the anterior wall of the choana. The jugal is not preserved on either side.

**Squamosal.** The right zygomatic process of the squamosal is broken, while it is well preserved on the left side, only missing its anterior tip. In dorsal view, the zygomatic process is slender, and its anterior end extends to the level of the choana in lateral view, just slightly posterior to the infratemporal crest. The supramastoid crest and lateral half of the glenoid process can be observed in dorsal view. The supramastoid crest is parallel to the lateral margin of the maxilla and is almost parallel to the anteroposterior axis. The squamosal fossa is dorsoventrally deep (7 mm) and laterally wide (23 mm), and it faces dorsally. In the lateral view, the zygomatic process is rectangular and anteriorly slightly projecting dorsally.

In ventral view, the mandibular fossa is large but shallow, and it faces medioventrally rather than anteroventrally. The post–tympanic process is slightly concaved. It is laterally bordered by the tympanosquamousal recess and anteriorly by the mandibular fossa. The posterior part of the postglenoid process is low. The tympanosquamosal recess is located immediately posteromedial to the mandibular fossa, and it is anteroposteriorly long and posteriorly wide. The falciform process is long and slender, and it makes the anterior wall of the cranial hiatus. In ventral view, the periotic fossa is rounded. The foramen ovale is preserved on each side of the alisphenoid, 5 mm anterior to the periotic fossa, and it is 3 mm in diameter. The squamosal is fused with the alisphenoid, exoccipital, and parietal.

**Supraoccipital.** Owing to the deformation of the occipitals, the dorsal part of the supraoccipital is compressed and skewed clockwise to the right side. The supraoccipital forms the nuchal crest and occupies the highest point of the cranium. Dorsally, the nuchal crest is thick and flat anteroposteriorly and laterally, and it is not extended anteriorly. In lateral view, the nuchal crest is appreciably higher than the maxilla and slightly overhangs anteriorly. In posterior view, the dorsal edge of the nuchal crest is circular and slightly rectangular.

In lateral view, the supraoccipital shield is flat and faces more posteriorly rather than dorsally. Therefore, the nuchal crest is short anteroposteriorly and not convex posteriorly in the dorsal view. Slightly lower than the nuchal crest, there is a deep depression at the center of the supraoccipital surface. The external occipital crest and suture between the supraoccipital and exoccipital are not present.

**Exoccipital.** The exoccipital is fused with the basioccipital and squamosal. In dorsal view, the occipital condyle is slightly separated from the cranium and pushed out posteriorly. This may be a result of deformation, because a crack on the exoccipital can be observed dorsally, laterally, and ventrally. In the posterior view, the foramen magnum is compressed dorsoventrally, and the left and right occipital condyles are turned clockwise. The surface of the occipital condyle is smooth, while the dorsal margin is damaged, and the presence or absence of a dorsal condyloid fossa is unclear. A short pedicle of the occipital condyle (approximately 15 mm in width) is developed, and the intercondyloid notch is U–shaped.

The jugular notch is deep in the ventral view, whereas the the presence or absence of hypoglossal foramen is unclear. The paroccipital process is well developed. It extends more ventrally than the basioccipital crest and laterally reaches the level of the lateral edge of the maxilla. The paroccipital concavity is shallow. The posterolateral sinus fossa is preserved but shallow.

**Basioccipital.** Both the left and right basioccipital crests are damaged at their posteroventral margins. In ventral view, it diverges posteriorly. The transverse width at the posterior end of the basioccipital crest is slightly wider than the occipital condyle. The basioccipital crest

contacts the posterior lamina of the pterygoid anteriorly. It covers most of the periotic fossa. The ventral surface of the basioccipital is flat and has no indication of muscular tubercle.

## Results

### Phylogeny

We found 52 most parsimonious trees with 3385 steps each. The strict consensus of these trees is shown in Fig 7. The 50% majority-rule consensus tree is also shown in S1 File. Our resulting trees show almost the same topology as was proposed by Guo and Kohno [8]; that is, all the 'kentriodontid' species that were identified by previous studies form a monophyletic clade, and were placed as the sister group to the Delphinoidea (Fig 7). Species of 'kentriodontid'

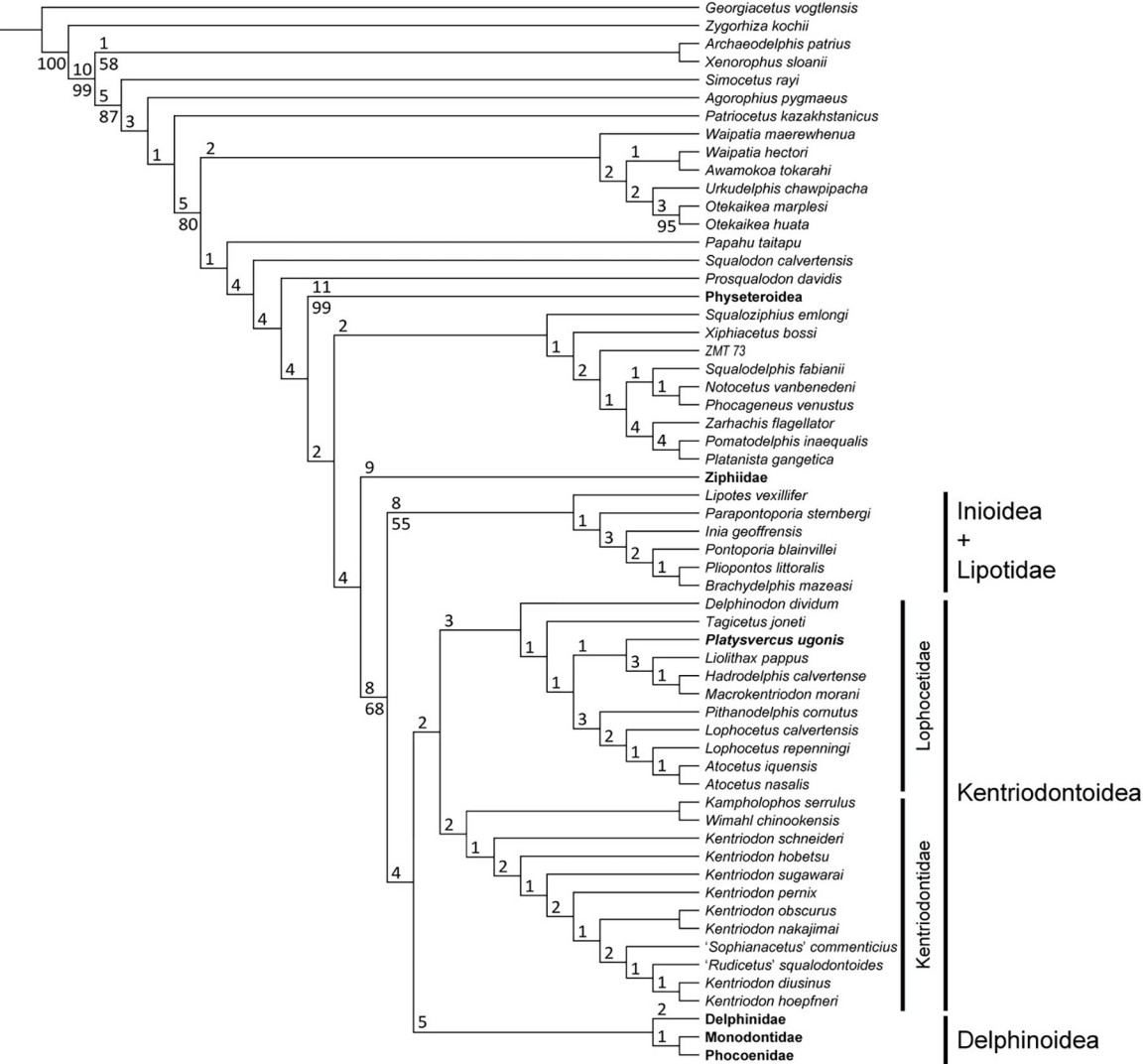

**Fig 7. Phylogenetic relationships of *Platysvercus ugonis* gen. et sp. nov.** Strict consensus tree resulting from 52 most parsimonious trees with tree constrained by the molecular trees of McGowen et al. [24–26], 3385 steps long, with the consistency index = 0.197 and the retention index = 0.568. Numbers above the nodes are Bremer support. Numbers below nodes indicate bootstrap values (1,000 replicated). The values less than 50% were omitted. The interspecific relationships of the Physeteroidea, the Ziphiidae, the Delphinidae, the Phocoenidae and the Monodontidae were omitted and these taxa were integrated into families/superfamilies.

dolphins were split into two sub–clades; i.e., the sub–clade that contains *Whimal*, *Kampholophos*, *Sophianacetus*, *Rudicetus* and *Kentriodon* (sub-clade A), and the sub–clade that contains *Delphinodon*, *Tagicetus*, *Macrokentriodon*, *Hadrodelphis*, *Liolithax*, *Atocetus* and *Lophocetus* (sub-clade B). UTHFM 00034 described here is included in the latter sub–clade and is located as a sister taxon to *Macrokentriodon*, *Hadrodelphis*, and *Liolithax*. Based on our phylogenetic analysis with species level comparisons above, we propose a new genus and species *Platysvercus ugonis* for UTHFM 00034 within the sub-clade B.

## Inter-specific comparisons in the Lophodontidae

Phylogenetic analysis shows that UTHFM 00034 is closely related to *H. calvertense*, *Li. Pappus* and *M. morani* and comprising a monophyletic group in the sub-clade B. UTHFM 00034 shares a synapomorphic character with these three species: narrow width of the premaxillae at the antorbital notches (character 65).

Atocetus iquensis, *A. nasalis*, *Lo calvertensis*, *Lo. repenningi*, and *Pi. cornutus* forms another monophyletic group in the Lophocetidae. UTHFM 00034 differs from this group in having the nasal located at nearly the same height as the frontals (character 126). In UTHFM 00034, the posterior end of the nasal process of premaxillae is faced anteriorly (character 111), and the nuchal crest is at the same level as the frontals (character 142). These features are different from *A. iquensis*, *Lo calvertensis*, *Lo. repenningi*, and *Pi. cornutus*. Also UTHFM 00034 differs from *A. nasalis*, *Lo calvertensis*, and *Pi. cornutus* in having the flat dorsal exposure of frontals (character 136). Furthermore, UTHFM 00034 differs from *T. joneti* with the posterolateral sulcus from premaxillary foramen extending to the anteroposterior level to the nasal openings (character 73), and dorsoventral thick anterior edge of the nasal (character 116). UTHFM 00034 is distinguishable from *D. dividum* with the maxillae exposed on the anterior edge of external bony nares (character 105), and depressed medial portions of the nasals (character 131).

The clade including UTHFM 00034, *H. calvertense*, *Li. Pappus* and *M. morani*, UTHFM 00034 differs from the other three species by having the followng synapomorphies: a small diameter of the largest functional tooth (character 33), the premaxillae raised towards the midline but concave transversely between the anteromedial sulci emanating from the premaxillary foramen (character 121), the posterior part of the suture between the nasals marked by a deep cleft (character 134), and the anterior margin of the pterygoid sinus fossa located posterior to the antorbital notch (character 190); also it differs from *H. calvertense* and *Li. Pappus* by having the following synapomorphies: the premaxillae is narrowing at the apex of the rostrum (character 4), and the premaxillary foramen is located laterally from the midpoint on the premaxilla (character 70).

## Paleobiogeography

The results of the ancestral distributional area reconstructions are shown in Fig 8 and S1 File. The ancestor of the clade Delphinida (i.e., kentriodontids *sensu lato* (*s.l.*) along with Delphinoidea, Inioidea and Lipotidae) was inferred to have emerged in the North Pacific Ocean (node 192, A, probability = 100%, shown in S1 File). The common ancestor of the monophyletic kentriodontids *s.l.* was also most likely originated with the sister clade, that is, the Delphinoidea, in the North Pacific Ocean (node 191, A, probability = 100%). However, the ancestral taxon of the Delphinoidea is unknown until the late Middle Miocene and is considered to have diversified just before the extinction of the kentriodontids *s.l.* based on fossil records (node 190, A, probability = 100%). After the emergence of the kentriodontids *s.l.* in the North Pacific, they spread out into the North Atlantic and became globally distributed in the Northern

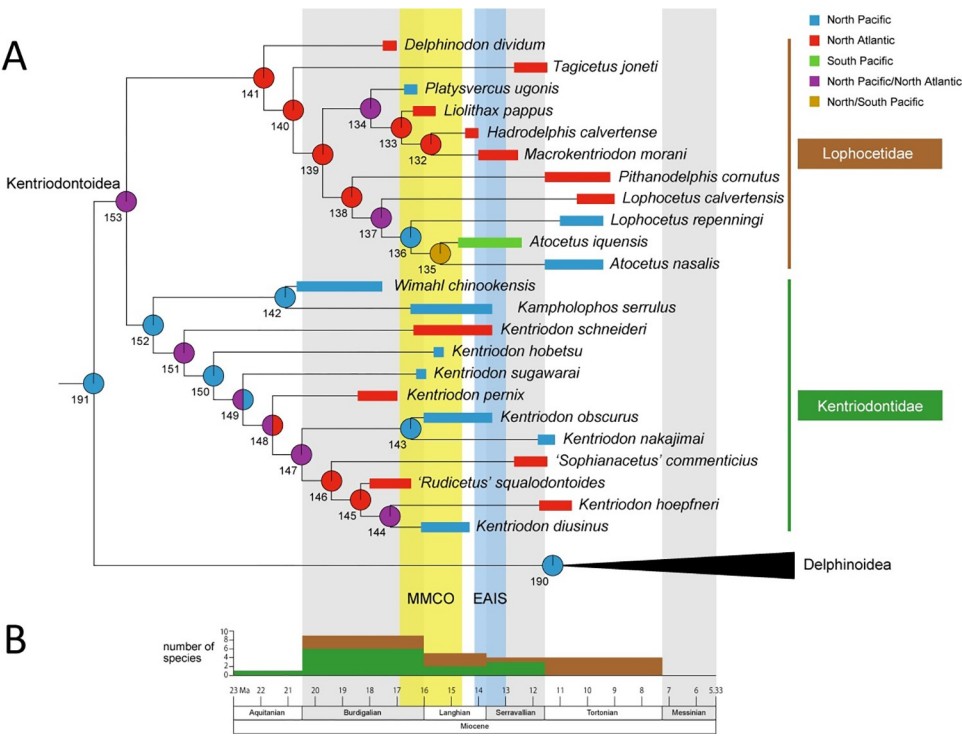

**Fig 8. Paleobiogeography and diversification of kentriodontoids.** (A) Summary of the topology, ancestral range reconstructions and paleobiogeography of the kentriodontoids. The topology of the Kentriodontoidea is based on our phylogenetic analysis (Fig 7). Stratigraphic ranges of each species are based on previous studies (see Supplemental Data 1 Table S1 and Appendix S5 in S1 File). Pie charts at each node show the ancestral range reconstructions, numbers below in the pie chart indicate node number (Supplemental Data 1 Fig S4 in S1 File for complete results). (B) Number of species richness based on first occurrences of each kentriodontoid at each stage in the Miocene. Species of the Lophocetidae are brown in color and those of the Kentriodontidae are dark green in color. Abbreviations: MMCO, the Middle Miocene Climatic Optimum; EAIS, expansion of the East Antarctic Ice Sheet.

Hemisphere (node 153, AB, probability = 100%). Afterward, the kentriodontids *s.l.* were split into two sub–clades, the 'North Pacific' clade and the 'North Atlantic' clade, respectively (node 152, A, probability = 100%; node 141, B, probability = 100%). This split into two sub–clades A and B is interpreted as vicariance, as will be discussed below. Similar vicariance events were also recognized just before or at the start of the Middle Miocene Climatic Optimum (MMCO) at approximately 17 Ma (Fig 8). These were represented by the North Pacific UTHFM 00034 and the North Atlantic *Liolithax pappus* within the 'North Atlantic' sub–clade and also by the North Atlantic *Kentriodon schneideri* and the North Pacific *K. hobetsu* in the 'North Pacific' sub–clade. After the second vicariance event, species within both sub–clades A and B diversified in the Northern Hemisphere, and the North Pacific genus *Atocetus* within the 'North Atlantic' sub–clade also extended into the South Pacific Ocean (node 135, AC, probability = 100%).

## Discussion

### Phylogeny and refined classification of 'kentriodontids'

All the kentriodontids *s.l.* known at least from crania are recognized as monophyletic, and 12 synapomorphies (i.e., characters 3, 7, 56, 62, 104, 140, 157, 225, 227, 278, 326, and 328) support this clade (see also the Diagnosis of Superfamily and S1 File). UTHFM 00034 described here is

recognized by three autapomorphies (i.e., characters 70, 121, and 190) within the kentriodon-tids *s.l.* (see also the Diagnosis of Genus and S1 File).

The monophyletic kentriodontids s.l. is placed as a sister group to the Delphinoidea, as has already been proposed by some earlier studies [5,6,8]. However, the taxonomic contents of the kentriodontids *s.l.* differ among them. Our analysis supported the results of Guo and Kohno [8], and the monophyletic kentriodontids *s.l.* include all of the species previously recognized as kentriodontids *s.l.* by 12 synapomorphies as was pointed out above. It is almost the same robustness as the superfamily Delphinoidea, which is supported by 18 synapomorphies. Also, bootstrap and Bremer support values (Fig 7) indicated the stability of this phylogenetic rela-tionship. In this regard, the kentriodontid clade could be recognized as a superfamilial rank as is the case for their sister taxon, the Delphinoidea. Hence, we propose the Superfamily Ken-triodontoidea, to include all the kentriodontids *s.l.*

Our phylogenetic analysis also indicates that the new clade Kentriodontoidea is divided into two sub–clades. The one includes *Kampholophos serrulus*, *Kentriodon diusinus*, *Ke. hobetsu*, *Ke. hoepfneri*, *Ke. nakajimai*, *Ke. pernix*, *Ke. schnederi*, *Ke. sugawarai*, *Rudicetus squa-lodontoides*, *Sophianacetus commenticius* and *Wimahl chinookensis*. Five synapomorphies (i.e., characters 183, 245, 264, 266, and 277) support this sub–clade (see also S1 File). Some earlier studies [4,42] included all the taxa of kentriodontines in the monohpyletic family Kentriodon-tidae, but Peredo et al. [5] suggested that the Kentriodontidae *sensu stricto* (*s.s.*) only included four genera of kentriodontids s.l. which is approximately identical to our definition of the Ken-triodontidae. Our analysis also confirmed their result and followed the family name for our resultant sub–clade as the Kentriodontidae *sensu* Peredo et al. [5] for this group.

The other one includes *Atocetus iquensis*, *Atocetus nasalis*, *Delphinodon dividum*, *Hadrodel-phis calvertense*, *Liolithax pappus*, *Lophocetus calvertensis*, *Lo. repenningi*, *Macrokentriodon morani*, *Pithanodelphis cornutus*, *Platysvercus ugonis* gen. et sp. nov. and *Tagicetus joneti*. Eight synapomorphies (i.e., characters 37, 70, 133, 134, 169, 258, 273, and 355) support this sub–clade (see also the Diagnosis of the Lophocetidae and S1 File). Given that this clade has as many synapomorphies as the family Kentriodontidae as well as the Delphinidae (three), Monodontidae (thirteen), and Phocoenidae (seven), this clade can also be recognized as a fam-ily within the Kentriodontoidea. Earlier studies like Muizon [3] and Barnes [2,42] assigned the whole kentriodontid-like species as the family Kentriodontidae, and subdivided into four sub-families; the Kentriodontinae, the Kampholophinae, the Lophocetinae and the Pithanodelphi-nae. Except for the monophyletic Kentriodontinae, all the members of the Lophocetinae and the Pithanodelphinae are included in the latter subclade, while the genera of the Kampholophi-nae are recognized as polyphyletic (*Kampholophos* is included in the Kentriodontidae and *Lio-lithax* is included in the latter subclade). Assigning a new family name from previously named subfamilies is more stable and helpful to future work, and the Lophocetinae Barnes, 1978 was proposed earlier and has priority between the two previously named subfamilies. Accordingly, we propose the Lophocetidae with a new definition for the name of this subclade.

The type genera and species of both families (i.e. *Kentriodon* for the Kentriodontidae and *Lophocetus* for the Lophocetidae) are represented by diagnostic specimens including the skull and tympanoperiotic. At present, these two families consist of only about 10 taxa respectively, which are fewer in comparison with other odontocete families, but the numbers of synapo-morphies at each node (five in the Kentriodontidae and eight in the Lophocetidae) confirm each clade as robust as the Delphinidae, Monodontidae, and Phocoenidae within the Delphi-noidea, as mentioned above. Again, the numbers of unique characters for each family are almost identical to that of other families within the Delphinida, but each kentriodontoid family cannot be defined by any unrivaled autapomorphic characters. Future investigations to iden-tify unique or autapomorphic characters will clarify the robustness of their relationships.

## Paleobiogeography of kentriodontoids

As has been suggested by some of the previous studies [4,43,44], kentriodontoids were recognized to have distributed worldwide during the Middle Miocene (Fig 8 and Supporting Information). The results of our study indicate that the common ancestor of the Kentriodontoidea emerged in the North Pacific Ocean and immediately spread out in the Northern Hemisphere. This initial diversification is interpreted to be the range expansion of their distribution. At the time of the first split of the ancestral kentriodontoids into two subgroups, one of which emerged in the North Pacific Ocean and became the ancestor of the Kentriodontidae. By contrast, another subgroup seems to have emerged in the North Atlantic Ocean and later evolved into the Lophocetidae. Both events occurred, at the latest, before the Middle Miocene as a vicariance event for the ancestral kentriodontoids (Fig 8A: nodes 152 and 141).

After the first emergence of the Kentriodontidae and the Lophocetidae, members of both families diverged sequentially and globally in the North Pacific and the North Atlantic through the late Early and the late Middle Miocene, respectively (Fig 8A and 8B: nodes 151, 147, 144, 137, 135, and 134). The Central American Seaway was opened intermittently during the Early and Middle Miocene [15,45], and it allowed repeated current connections as global ocean circulation [16,46]. This meant that marine vertebrates were able to migrate into and/or between the Pacific Ocean and the Atlantic Ocean without formidable barriers as a land bridge between North and South America [47–49]. In addition, the dispersal of the Kentriodontidae is thought to be somewhat earlier than that of the Lophocetidae based on fossil records (Fig 8B and Supporting Information). The first occurrences (or the oldest known records) of each species within the Kentriodontidae during the Miocene are summarised as follows: one taxon appeared in the Aquitanian, six taxa in the Burdigalian, two taxa in the Langhian, and three taxa in the Serravallian. On the other hand, the first occurrences (or the oldest known records) of species within the Lophocetidae are as follows: three taxa in the Burdigalian, three taxa in the Langhian, one taxon in the Serravallian, and four taxa in the Tortonian. Both families of the Kentriodontoidea declined and became extinct in the Tortonian (Fig 8A and 8B: nodes 144 and 138). Further analysis could improve the current reconstruction of their paleobiogeography when even more specimens could be added to the cladogram. Recently, many specimens of delphinidans including kentriodontoids have been reported such as those undescribed specimens discovered in Peru [50,51].

Several previous studies have suggested that the diversification and/or extinction events that occurred in the Miocene were the result of global ocean climate changes [18,52–54], based on carbon and oxygen isotopic changes [14], which were related to Miocene ice–sheet expansion events [17,55,56]. Other previous research [57] also suggests that the sea surface temperatures decreased by about 4˚C –7˚C in the North Pacific Ocean and the North Atlantic Ocean from the Middle Miocene Climatic Optimum (MMCO, 16.9–14.7 Ma) to the beginning of the expansion of the East Antarctic Ice Sheet (EAIS, 14–13 Ma). However, the interaction between the diversification and extinction of kentriodontoids and these climatic events has not yet been proven to be the cause of the extinction of marine mammals. Nevertheless, the vicissitudes of Kentriodontoidea were obviously influenced by these climate events. That is, nine taxa emerged within the MMCO, (16.9–14.7 Ma), and seven taxa were extinct around the period of or just after EAIS (Fig 8A). These climatic events may not only be the cause of the rise and fall of the kentriodontoids, but may also be related to the turnover event with the delphinoids around the Serravallian ([58], Plate 16a). However, it is still difficult to reveal the function and response between global climate change and these faunal turnover events.

The faunal turnover events for the Kentriodontoidea must also be considered in a different perspective, especially in the context of their extinction. The oldest known record of their sister

taxon, the family Delphinoidea, goes back to 11.29–11.25 Ma [59], in the Tortonian of the Middle Miocene. Previous studies [12,13] have suggested that the origin and early diversification of the Delphinoidea were recognized in the Middle to Late Miocene. Although Delphinoidea have been considered to have the same ecological niches as environmental preferences [10] and feeding strategies [11] with most of the Kentriodontoidea, the absence of the Delphinoidea in the Middle Miocene might have been the result of niche partitioning based on our phylogenetic and paleobiogeographic analysis. Similar to some research [4], the results suggest that the Kentriodontoidea might have declined in their niche by the diversification of the Delphinoidea in the Late Miocene. This also indicates that the early diversification of the Delphinoidea may have occurred as a result of this turnover event.

## Conclusions

An extinct delphinidan *Platysvercus ugonis* was described as a new genus and species belonging in the Kentriodontoidea, new rank, based on a nearly complete skull, UTHFM 00034, from the upper Lower Miocene Sugota Formation, Akita Prefecture, northern Japan. *Platysvercus ugonis* gen. et sp. nov. shares the kentriodontoid synapomorphies, but it is distinguishable from other kentriodontoids by a unique combination of the following characters: the premaxillary foramen is located at the midpoint of the premaxilla; the premaxillae are raised up toward the midline and concave transversely, and the anterior level of the pterygoid sinus fossa is interrupted at the portion posterior to the antorbital notch. Based on our phylogenetic analysis, the systematics of the kentriodontoids were refined: i.e., the superfamily Kentriodontoidea mentioned above is proposed for all the species formerly included in the kentriodontids *s.l.* The new superfamily is subdivided into two families, the Kentriodontidae *sensu* Peredo et al. (2018) and the family Lophocetidae new rank with new definition. *Platysvercus ugonis* gen. et sp. nov. is nested in this new family Lophocetidae.

Our paleobiogeographic analysis of the new superfamily Kentriodontoidea indicates that their common ancestor emerged in the North Pacific Ocean and immediately spread out in the Northern Hemisphere. The initial diversification of the ancestral taxon or taxa of kentriodontoids is interpreted to be a result of the range expansion of their distribution in the Northern Hemisphere, and the Kentriodontidae might have emerged and diversified in the North Pacific Ocean, while the Lophocetidae fam. nov. might have emerged and diversified in the North Atlantic Ocean as a vicariance event for the kentriodontoids. However, the kentriodontoids decreased and became extinct following the climatic change during the Middle Miocene, synchronously influenced by the expansion of the East Antarctic Ice Sheet. The Kentriodontoidea also seem to have declined in their niche due to the rapid diversification of the Delphinoidea during the Middle to Late Miocene boundary age as a turnover event around the Serravallian.

## Supporting information

**S1 File. Phylogenetic trees, list of taxa and characters states and materials and results of the ancestral range reconstructions.**
(PDF)

**S2 File. Data matrix used for the phylogenetic analysis in ".nex" format.**
(NEX)

**S3 File. Data matrix used for the phylogenetic analysis in ".tnt" format.**
(TNT)

**S4 File. README file including all the steps of the phylogenetic analysis.**
(OUT)

**S5 File. Tree file from phylogenetic analysis.**
(TRE)

## Acknowledgments

We wish to thank T. Sato for discovering the holotype skull of *Platysvercus ugonis* gen. et sp. nov. described here for the UTHFM. We also thank K. Ito, F. Kakizaki, and S. Goto (both UTHFM) for permitting us to describe UTHFM 00034. We would like to thank Y. Tajima (NMNS) and T. Kimura (GMNH) for providing access to the collections under their care. We also thank Y. Tajima, H. Ichishima (Fukui Prefectural Dinosaur Museum), Y. Tanaka (Osaka Museum of Natural History), and T.K. Yamada (NMNS) for providing useful advice. We are grateful to Y. Hasegawa (formerly Yokohama National University, now Iida City Museum), K. Sashida (formerly University of Tsukuba, now Mahidol University), S. Agematsu (Univ. Tsukuba) and K. Tanaka (Univ. Tsukuba), and I. Tanaka (formerly Univ. Tsukuba, now the Geological Survey of Japan, AIST) and Y. Shigeta (NMNS/Univ. Tsukuba) for providing useful advice, discussion, and generous encouragement during the course of this study.

## Author Contributions

**Conceptualization:** Zixuan Guo, Naoki Kohno.

**Data curation:** Naoki Kohno.

**Formal analysis:** Zixuan Guo.

**Software:** Zixuan Guo.

**Supervision:** Naoki Kohno.

**Validation:** Naoki Kohno.

**Visualization:** Zixuan Guo.

**Writing – original draft:** Zixuan Guo, Naoki Kohno.

**Writing – review & editing:** Zixuan Guo, Naoki Kohno.

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
