## [Decision Letter · Decision Letter 0]

12 Jul 2022

PONE-D-21-33588

An early Miocene kentriodontoid (Cetacea: Odontoceti) from the western North Pacific, and its implications for their phylogeny and paleobiogeography

PLOS ONE

Dear Dr. Guo,

Thank you for submitting your manuscript to PLOS ONE. After careful consideration, we feel that it has merit but does not fully meet PLOS ONE’s publication criteria as it currently stands. Therefore, we invite you to submit a revised version of the manuscript that addresses the points raised during the review process.

We look forward to receiving your revised manuscript.

Kind regards,

Luca Pandolfi

Academic Editor

PLOS ONE

Journal Requirements:

2. In your manuscript, please provide additional information regarding the specimens used in your study. Ensure that you have reported specimen numbers and complete repository information, including museum name and geographic location. 

For more information on PLOS ONE's requirements for paleontology and archaeology research, see https://journals.plos.org/plosone/s/submission-guidelines#loc-paleontology-and-archaeology-research.

4. We note that Figure 1 in your submission contain map images which may be copyrighted. All PLOS content is published under the Creative Commons Attribution License (CC BY 4.0), which means that the manuscript, images, and Supporting Information files will be freely available online, and any third party is permitted to access, download, copy, distribute, and use these materials in any way, even commercially, with proper attribution. For these reasons, we cannot publish previously copyrighted maps or satellite images created using proprietary data, such as Google software (Google Maps, Street View, and Earth). For more information, see our copyright guidelines: http://journals.plos.org/plosone/s/licenses-and-copyright.

5. Please take this opportunity to be sure you have met all of our guidelines for new species. For proper registration of a new zoological taxon, we require two specific statements to be included in your manuscript.

A. In the Results section, the globally unique identifier (GUID), currently in the form of a Life Science Identifier (LSID), should be listed under the new species name, for example:

Anochetus boltoni Fisher sp. nov. urn:lsid:zoobank.org:act:B6C072CF-1CA6-40C7-8396-534E91EF7FBB

Another LSID for the manuscript itself should also appear within the Nomenclature statement. You will need to contact Zoobank (zoobank.org/About) to obtain a GUID (LSID). You should receive one LSID for your manuscript and a separate, unique LSID for the new species. 

B. Please also insert the following text into the Methods section, in a sub-section to be called ""Nomenclatural Acts"":

The electronic edition of this article conforms to the requirements of the amended International Code of Zoological Nomenclature, and hence the new names contained herein are available under that Code from the electronic edition of this article. This published work and the nomenclatural acts it contains have been registered in ZooBank, the online registration system for the ICZN. The ZooBank LSIDs (Life Science Identifiers) can be resolved and the associated information viewed through any standard web browser by appending the LSID to the prefix """" ext-link-type="uri" xlink:type="simple">http://zoobank.org/"". The LSID for this publication is: urn:lsid:zoobank.org:pub: XXXXXXX. The electronic edition of this work was published in a journal with an ISSN, and has been archived and is available from the following digital repositories: PubMed Central, LOCKSS [author to insert any additional repositories].

All PLOS ONE articles are deposited in PubMed Central and LOCKSS. If your institute, or those of your co-authors, has its own repository, we recommend that you also deposit the published online article there and include the name in your article.

Following a recent ruling by the International Commission on Zoological Nomenclature, electronic journals are now a valid format for publication of new zoological taxa. In order to ensure the valid publication of your new species, please be sure to include the updated version of Nomenclatural Acts (above). A complete explanation of our guidelines for publishing new species can be found on our website: http://www.plosone.org/static/guidelines#zoological.

Additional Editor Comments (if provided):

Dear Author,

two reviewers revised your paper and added some suggestions and comments that greatly will improve your manuscript. Please take into account all the suggestions, in particular those of reviewer2 and please submit a point-by-point reply for each comment.

I hope to read soon your revised version.

kind regards

Luca Pandolfi

Reviewers' comments:

Reviewer's Responses to Questions

**Comments to the Author**

1. Is the manuscript technically sound, and do the data support the conclusions?

Reviewer #1: Partly

Reviewer #2: Partly

2. Has the statistical analysis been performed appropriately and rigorously? 

Reviewer #1: Yes

Reviewer #2: Yes

3. Have the authors made all data underlying the findings in their manuscript fully available?

Reviewer #1: Yes

Reviewer #2: Yes

4. Is the manuscript presented in an intelligible fashion and written in standard English?

Reviewer #1: Yes

Reviewer #2: Yes

5. Review Comments to the Author

Reviewer #1: Your study makes 3 important contributions. It describes a new species of Kentriodontidae that has clear affinities with Liolithax and Lophocetus, you conduct a large phylogenetic analysis of odontocetes including most kentriodontidae, and you investigate the biogeographic history of this group. I have attached a copy of your ms with detailed comments and suggestions for improvements. Below are what I consider to be the most important issues to address:

1) Taxonomic revisions: I am supportive of naming new groups to further discussions on Kentriodontid phylogeny. However, I strongly urge you to define your clades as well as diagnose them. This is important because further studies will likely find slightly different relationships, and definitions will help future authors apply your names to their trees. Phylogenetic definitions are, in my opinion, most helpful, and Paul Sereno has published several papers how to define such clades so that they are stable.

You should also describe more of the existing literature concerning named taxa for clades within Kentriodontidae. For example de Muizon 1988 has a phylogeny of kentriodontids and uses the subfamilies Kentriodontinae, Lophocetinae, and Pithanodelphinae. You should discuss more how your results correspond to this earlier work.

Finally I think you need to explain more your reasoning on why a new superfamily is needed. Historically the clade you name Kentriodontoidea has been referred to as Kentriodontidae. In the ms you place kentriodontids in quotes, which usually indicates a paraphyletic assemblage. However your study actually supports its monophyly. I am assuming you did this because you are distinguishing the traditional definition from your new one. I would also encourage you to revise the diagnosis for Kentriodontidae in this paper, in addition to your new family Delphinodontidae.

2) In the methods section you need to detail how you conducted searches used to estimate Bremer support and bootstrap values.

3) I am not familiar with the program used for ancestral state reconstructions, but it seems important to describe how you (or the program) differentiates vicariance from dispersal. I am not saying that vicariance is impossible in the marine environment, but seems like dispersal might be more common.

4) In the description it is unclear how your are using the term presphenoid and how it might relate to the ethmoid. I suggest you clarify your meaning and refer to Ichishima 2016 for the most recent discussion of the homology of bones in this region.

5) In File S1 Fig. S2 looks corrupted, or maybe there are two trees, ones pasted over the other?

Best,

Jonathan Geisler

Reviewer #2: Dear editors and authors,

please excuse my tardy reply, and thank you very much for your patience in waiting for my review.

I have read over the above paper and support its publication on PLOS ONE after suitable moderate revisions.

Here you are my main comments:

- Though mostly clear, the English text should be revised. I have included some possible edits in the attached revised manuscript file. That said, as I am not a native English speaker, I do not feel much qualified to help the authors on this very issue.

- Comparisons are completey absent from the Description section. Considering also that the diagnosis is not comparative, comparisons appear to be completely absent from the whole paper. This must be fixed before acceptance: detailed comparisons may be either interspersed within the description, or concentrated in a devoted paragraph at the end of the Systematics section. Anyway, including anatomical comparisons is mandatory in a paper that describes a new genus a species (otherwise, the legitimacy of the newly instituted taxon may even be disputed - in other words, it may not be obvious why Platysvercus ugonis should represent a new taxon).

- I am not sure whether splitting the kentriodontids into two families is a good idea. After all, there are recent phylogenies (e.g. those by Kimura Hasegawa (2019) and Bianucci et al. (2022)) in which such clades are not recovered. Whereas there's more or less consensus on the existence of a group of kentriodontids "sensu stricto" including mostly Kentriodon species, the same cannot be said for the "delphinodontids". In addition, and closely related to the previous point, I wonder whether naming the new family after Delphinodon is a good idea. After all, Delphinodon is rather close to Kentriodon in morphology and size; furthermore, it is recovered as the basalmost representative of the "delphinodontids" in your analysis, as well as the earliest branching representative of Delphinida(!!) in the aforementioned papers by Kimura Hasegawa (2019) and Bianucci et al. (2022).

- If you decide for keeping the new family Delphinodontidae, redefining the family Kentriodontidae with a new diagnosis would be very helpful here, and much in line with the "spirit" of the Principle of Coordination.

- I think that the palaeobiogeographic paragraph of the Discussion ignores the occurrence of Kentriodon sp. in the Lower Miocene of Peru. Please discuss the data on Kentriodon presented in the papers by Bianucci et al. (2018) and Di Celma et al. (2019). In addition, the Middle and Late Miocene of the Pisco Formation of Peru includes many kentriodontid-like specimens (Collareta et al., 2021).

- A more general reflection: the fossil record is certainly intrinsically fragmentary... However, Bianucci et al.'s phylogeny (in the supp. mat.), in which several kentriodontid-like dolphins (including many "delphinodontids") are more derived than (and not sister group to) kentriodontids s.s. avoids the need for a ghost lineage of basal delphinoids ranging from the Aquitanian/Burdigalian transition through the Burdigalian and Middle Miocene well into the Tortonian. I think you should consider (=mention and discuss) the alternative phylogenetic reconstruction by the recent paper by Bianucci et al. (2022).

Minor comments are as per the attached edited manuscript file.

Best wishes,

Alberto Collareta

Bianucci et al. (2018).- Taphonomy and palaeoecology of the lower Miocene marine vertebrate assemblage of Ullujaya (Chilcatay Formation, East Pisco Basin, southern Peru).- Palaeogeography, Palaeoclimatology, Palaeoecology, vol. 511, p. 256-279.

Bianucci et al. (2022).- The origins of the killer whale ecomorph.- Current Biology, vol. 32, p. 1843-1851.

Collareta et al. (2021).- Vertebrate Palaeoecology of the Pisco Formation (miocene, Peru): glimpses into the ancient humboldt Current ecosystem.- Journal of Marine Science and Engineering, vol. 9, article #1188.

Di Celma et al. (2019).- Allostratigraphy and paleontology of the lower Miocene Chilcatay Formation in the Zamaca area, East Pisco basin, southern Peru.- Journal of Maps, vol. 15, p. 393-405.

Kimura Hasegawa (2019).- A new species of Kentriodon (Cetacea, Odontoceti, Kentriodontidae) from the Miocene of Japan.- Journal of Vertebrate Paleontology, vol. 39, article #e1566739.

6. PLOS authors have the option to publish the peer review history of their article (what does this mean?). If published, this will include your full peer review and any attached files.

Reviewer #1: No

Reviewer #2: No

---

## [Author Response · Author response to Decision Letter 0]

10 Sep 2022

First of all, we deeply thank all the reviewers for their generous comments and suggestions for our manuscript. We thoroughly revised the manuscript with additional data following recommendations and suggestions from reviewers and editors. Then, our revised manuscript was greatly improved. Changes are recorded with the track and change function in the revised manuscript. We also submit the clean version of the revised manuscript after the tracked change copy. We responsed all the reviewers’ and editor's comments, which are included in the rebuttal letter. We hope that our revision is acceptable for publication.

---

## [Decision Letter · Decision Letter 1]

18 Oct 2022

PONE-D-21-33588R1

An early Miocene kentriodontoid (Cetacea: Odontoceti) from the western North Pacific, and its implications for their phylogeny and paleobiogeography

PLOS ONE

Dear Dr. Guo,

Thank you for submitting your manuscript to PLOS ONE. After careful consideration, we feel that it has merit but does not fully meet PLOS ONE’s publication criteria as it currently stands. Therefore, we invite you to submit a revised version of the manuscript that addresses the points raised during the review process.

Dear Authors,

Thank you for your revised version and for the changes you provided on the manuscript.

The revised version has been checked by two indipendent reviewers and both suggested minor revision.

Please address these last comments and observations before to submit a final version of the manuscript

Sincerely yours

Luca Pandolfi

If applicable, we recommend that you deposit your laboratory protocols in protocols.io to enhance the reproducibility of your results. Protocols.io assigns your protocol its own identifier (DOI) so that it can be cited independently in the future. For instructions see: https://journals.plos.org/plosone/s/submission-guidelines#loc-laboratory-protocols. Additionally, PLOS ONE offers an option for publishing peer-reviewed Lab Protocol articles, which describe protocols hosted on protocols.io. Read more information on sharing protocols at https://plos.org/protocols?utm_medium=editorial-emailutm_source=authorlettersutm_campaign=protocols.

We look forward to receiving your revised manuscript.

Kind regards,

Luca Pandolfi

Academic Editor

PLOS ONE

Journal Requirements:

Reviewers' comments:

Reviewer's Responses to Questions

**Comments to the Author**

1. If the authors have adequately addressed your comments raised in a previous round of review and you feel that this manuscript is now acceptable for publication, you may indicate that here to bypass the “Comments to the Author” section, enter your conflict of interest statement in the “Confidential to Editor” section, and submit your "Accept" recommendation.

Reviewer #2: (No Response)

Reviewer #3: (No Response)

2. Is the manuscript technically sound, and do the data support the conclusions?

Reviewer #2: Yes

Reviewer #3: Yes

3. Has the statistical analysis been performed appropriately and rigorously? 

Reviewer #2: Yes

Reviewer #3: N/A

4. Have the authors made all data underlying the findings in their manuscript fully available?

Reviewer #2: Yes

Reviewer #3: Yes

5. Is the manuscript presented in an intelligible fashion and written in standard English?

Reviewer #2: Yes

Reviewer #3: Yes

6. Review Comments to the Author

Reviewer #2: Dear Editors and Authors,

I have read over the revised version of the paper by colleagues Guo Kohno (and especially their responses to the reviewers' comments). It is clear that most of the reviewers' comments have been carefully handled by the authors, resulting in a strengthened manuscript.

- The authors have addressed my previous comment on the need for detailed osteoanatomic comparisons by adding a paragraph entitled “Inter-specific comparisons in the Lophodontidae” to the Discussion. Considering the final sentence of this paragraph (“Based on our phylogenetic analysis with species level comparisons above, we propose a new genus and species Platysvercus ugonis for UTHFM 00034 within the Lophocetidae”), the whole paragraph may rather be placed within the Results. Such a change may need to be accompanied by some local rewriting.

- I understand why the authors are reluctant to redefine Kentriodontidae. However, as also suggested by the other reviewer, recalling the diagnosis of this family (as retrieved in the present work, and sensu Peredo) would surely prove useful.

- My main concern with this manuscript still regards the English text. Though the majority of the manuscript is clear, not all the additions to the text have been ameliorative of this aspect. However, I am not a native English speaker, and as such, I am not particularly qualified to comment on this very issue.

I would suggest you to consider the above points before proceeding with publishing the paper.

Congratulations to the authors for this nice contribution to marine mammal palaeontology.

Best wishes,

Alberto Collareta

Reviewer #3: Dear authors,

I have been assigned to review this manuscript without being involved in the first round of reviews (of which I had access, along with your replies). The manuscript is sound and contains important new information regarding a new species of kentriodontid. The taxon sampling is great (it includes many kentriodontids) and the phylogenetic analysis is adequate. I have never performed a biogeographic analysis as done here using S-DIVA and therefore my expertise falls out of it, but I would like to share some general concerns that I hope are relevant and helpful below.

General comments:

Taxonomy

I understand the basis for suggesting a new rank for the group traditionally known as Kentriodontidae, namely the finding in this study of two subclades: (A) containing Kentriodon and related taxa (in the manuscript referred to “Kentriodontidae” or “Kentriodontidae sensu stricto”); and (B) a subclade containing taxa originally assigned to Kentriodontinae (most notably, Delphinodon dividum) together with taxa assigned to the traditionally-known subfamilies Lophocetinae and Pithanodelphinae (in this version of the manuscript collectively referred to as Lophocetidae). The assemblage of taxa in subclade B does not allow the use of the terms Lophocetinae and Pithanodelphinae in the traditional sense anymore.

Despite this, I think that the use of new rankings (Kentriodontoidea, and within this clade Kentriodontidae and Lophocetidae) does not help and in fact brings more confusion. A good example is in this manuscript: to differentiate between the “old” (traditional) term Kentriodontidae and the new term Kentriodontidae, the authors chose to refer to the old term Kentriodontidae using quotation marks and the informal name (“kentriodontids”) in the abstract and introduction, but this generally implies the group is paraphyletic or polyphyletic (as mentioned in the previous round of reviews). In this manuscript, it means indeed a monophyletic grouping that is used as a placeholder for what later would be defined as Kentriodontoidea. In the discussion we also find a new term “Kentriodontidae sensu lato” that somewhat is a synonym of the “kentriodontids” of the abstract and introduction and Kentriodontoidea of the title, the systematic section, and some other parts of the text. The use of so many names for the same clade throughout the manuscript is confusing and, in my view, does not help to better understand the message nor helps future papers probably dealing with similar issues.

I strongly encourage you to keep the traditional definition of family Kentriodontidae and to avoid any re-rankings. The subclades can be referred to using letters (as I did in the previous paragraph) or alternatively as the kentriodontinae and the non-kentriodontinae clade; both provide a quick aid to navigate through the information and avoid adding to the already convoluted taxonomy of this group.

Biogeography

Insights into chronology and distribution of kentriodontids are most needed, particularly because they may help to disentangle their complex taxonomic history, which is far from settled. I particularly like the part of the discussion where climate change and faunal turnovers are put into context. I think that the methodology section is missing some relevant details: in the methods the data was apparently assembled into 7 regions (A–G), but in Figure 8 there are some new categories: North Pacific/North Atlantic and North/South Pacific; in supplement 1, Figure S4 I see almost 40 categories depicted. I understand these are ranges, and some cetaceans are cosmopolitan and others have a very restricted distribution, however, I do not understand the need of pie charts and the reporting of probabilities in the results then, because almost all the pies are uniform in color and the probabilities reported are frequently 100%. Refer for example to Figure 8: the clade that includes the new taxon Platysvercus ugonis is phylogenetically reconstructed along with Liolithax, Hadrodelphis and Macrokentriodon (all 3 from the North Atlantic), further, Delphinodon and Tagicetus (the closest stemward relatives) are also from the North Atlantic. If the purple category (North Pacific/North Atlantic) did not exist, I would expect the pie in node 134 to show ¾ of red (indicating that the origin was most likely in the North Atlantic and Platysvercus is an immigrant in the North Pacific (there are probably more complex ways of seeing this, with varying percentages). A very simplistic a discrete character state reconstruction (reconstructing distribution as a state under parsimony) would again probably place a 100% red color in the node 134. In short: I do not understand why there are categories that lump areas (as the purple and yellow-brown categories in Figure 8), they seem counterintuitive to me, but again, I cannot really bring more constructive comments here.

Specific comments, number referring to line number in the clean copy (the one lacking track changes):

94-which consensus tree, majority rule or strict?

Table 1- check major and minor diameter of temporal fossa proper, are the values swapped?

451-and Lipotidae?

452-node 192 not shown in Figure 8, maybe remove from the main results as it is show only in supplement?

601602-strange wording, maybe “could improve the current…”?

634- either these turnover events or this turnover event

644- …of the kentriodontids were refined

657- the kentriodontids (no caps) decreased… in diversity?

Reference 51 – Capitalize Pisco Formation, Peru, and Humboldt

Figure 2B-“preorbital process” is showing the postorbital process; there is a missing arrow for the supraoccipital; arrow pointing the posterolateral sulcus is a few mm too short and not showing the sulcus; the “presphenoid” is in my interpretation the ethmoid.

Figure 3B, Figure 4B (and description of supraoccipital)- the “pedicle of occipital condyle” is a term I have never seen before and it is not in Mead and Fordyce’s (2009) lexicon. Is it relevant to have? Could you provide a reference on who defined that term?

Figure 4B- I interpret the “posterior lamina of pterygoid” as the medial lamina of the pterygoid of the pharyngeal crest of Mead and Fordyce (2009).

Kind regards,

Gabriel Aguirre-Fernández

7. PLOS authors have the option to publish the peer review history of their article (what does this mean?). If published, this will include your full peer review and any attached files.

Reviewer #2: No

Reviewer #3: **Yes: **Gabriel Aguirre-Fernández

---

## [Author Response · Author response to Decision Letter 1]

16 Dec 2022

First of all, we deeply thank all the reviewers for their generous comments and suggestions for our manuscript. We thoroughly revised the manuscript with additional data following recommendations and suggestions from reviewers and editors. Then, our revised manuscript was greatly improved. Changes are recorded with the track and change function in the revised manuscript. We also submit the clean version of the revised manuscript after the tracked change copy. We responsed all the reviewers’ and editor's comments, which are included in the rebuttal letter. We hope that our revision is acceptable for publication.

---

## [Editor Report · Decision Letter 2]

23 Dec 2022

An Early Miocene kentriodontoid (Cetacea: Odontoceti) from the western North Pacific, and its implications for their phylogeny and paleobiogeography

PONE-D-21-33588R2

Dear Dr. Guo,

We’re pleased to inform you that your manuscript has been judged scientifically suitable for publication and will be formally accepted for publication once it meets all outstanding technical requirements.

Kind regards,

Luca Pandolfi

Academic Editor

PLOS ONE
---

## [Editor Report · Acceptance letter]

11 Jan 2023

PONE-D-21-33588R2 

An Early Miocene kentriodontoid (Cetacea: Odontoceti) from the western North Pacific, and its implications for their phylogeny and paleobiogeography 

Dear Dr. Guo:

I'm pleased to inform you that your manuscript has been deemed suitable for publication in PLOS ONE. Congratulations! Your manuscript is now with our production department. 

Kind regards, 

on behalf of

Dr. Luca Pandolfi 

Academic Editor

PLOS ONE